# On Connection between `CLS` Token and Virtual Node: Are they both sides of the same coin?

## Abstract

Transformers emerged as a promising tool in learning long-range dependencies in language tasks. Recently, it has been shown that the self-attention module in the Transformer operates on the complete graph of tokens obtained from the input dataset. Specially, in language and vision-related tasks, a `CLS` token is added at the beginning of the token sequence to extract the representation of the entire sequence. On the other hand, a Virtual Node (VN) is added in a graph to mitigate the effect of Oversquashing, an information bottleneck typically observed in Graph Neural Networks (GNNs). In this work, we observe that both `CLS` token and VN are structurally identical in their respective graphs. Both `CLS` token and VN are connected to every other token or node and aggregate features from all other remaining tokens or nodes in a similar way. Although the embedding of the `CLS` token is used to classify the input sequence, the embedding of VN is not fully explored; instead, standard pooling is employed to extract the graph-level representation for classification. Thus, to bridge the gap, we consider representations of VNs to solve classification tasks on standard graph datasets and observe surprisingly competitive performances.

## 1 Introduction

Transformers (Vaswani et al., 2017) are the cornerstone of the Large Language Models (LLMs) and modern deep learning. The meteoric rise of Transformers reshaped the trajectory of long-context learning in language tasks. The heart of the Transformer architecture contains the self-attention module that computes dot-product attention between the tokens that constitute the sequence. The estimated attention scores are employed to update the token embeddings. Recently, (Joshi, 2025) establishes a compelling connection between Transformer and Graph Neural Network (Scarselli et al., 2008; Kipf & Welling, 2016). They demonstrate that applying a Transformer on a sequence of tokens is equivalent to estimating self-attention on a fully connected graph of tokens. Further, the work claims that the attention module is formally equivalent to the attention estimation strategy in GAT (Veličković et al., 2017).

Notably, the impact of the Transformer can also be witnessed in the domains of image and graph data. Vision Transformer (ViT) (Dosovitskiy et al., 2020) creates image patches, and a sequence of patches is fed into the Transformer. In the case of graphs, we consider nodes, edges, or subgraphs as the tokens for the input to Graph Transformer (GT) (Dwivedi & Bresson, 2020). For language and vision-related tasks, a new token is added at the beginning of the input token sequence, commonly known as `CLS`. After propagating through the Transformer layers, the embedding of the `CLS` token is treated as input to the classification head. Note that `CLS` embedding is the result of the aggregated features across the rest of the tokens.

On a different note, the addition of Virtual Node (VN) is an effective approach employed to mitigate the effects of Oversquashing (Alon & Yahav, 2020). The oversquashing is an information bottleneck that arises when an exponential amount of information needs to be compressed into fixed-dimensional feature vectors. The addition of VN means that a new node is added to the underlying graph, which is connected to every other node present in the graph. The feature initialization of VN can be random, and representation updates occur by propagating through message passing layers. Observe that a single VN reduces the maximum shortest path distance (i.e., diameter of the graph) to 2 between any pair of nodes. Specifically, every node

is connected to every other node via the newly added virtual node, which results in unhindered message passing.

**In this work**, we identify a structural similarity between `CLS` token and Virtual Node within the framework of the complete graph over the tokens in Transformers and nodes in an input graph. Despite their distinct purposes, both mechanisms update their representations through feature aggregation either from remaining tokens or nodes. `CLS` token is explicitly designed to aggregate information from the tokens derived from the source data, resulting in an embedding that captures the characteristics of the entire token set. Conversely, VNs are primarily incorporated to enable unrestricted message propagation throughout the graph topology. Earlier, we observed the structural correspondence between `CLS` and VN, respectively, in the token graph and input graph. At this stage, we investigate whether embeddings obtained from VN can effectively contribute to graph classification performance. To address the gap, we conduct a set of experiments to provide embeddings of VNs to the classifier head. We employ three distinct initializations to the embeddings of VNs and study their effects individually. Our results demonstrate that this approach achieves performance comparable to established graph pooling techniques, including Mean, Max, and Sum pooling methods.

**Contribution.** Our key contributions are (1) we are the first to observe the structural correspondence between `CLS` token and Virtual Node in the context of the complete graph constructed over the tokens. Both are directly connected to the rest of the tokens or nodes in their respective setups. (2) Despite their different utility, we consider the representation of VN in solving graph classification as similar to `CLS` token in Transformers. We observe the competitive performance when we consider the embedding of VN compared to other standard pooling techniques, establishing a deeper connection between `CLS` token and VN.

## 2 Preliminaries

### 2.1 Transformer

Consider the $t$-length sequence of input tokens $\mathbf{S} = \{s_1, s_2, \cdots, s_t\}$. The token embeddings are presented as $\mathbf{X} = [x_1, x_2, \cdots, x_t] \in \mathbb{R}^{t \times d}$ where each token $s_i$ is equipped with an embedding $x_i$ of $d$-dimensional vector. Following the notion from (Vaswani et al., 2017), we transformed token embeddings into Query $\mathbf{Q}$, Key $\mathbf{K}$, and Value $\mathbf{V}$ matrices as follows,

$$\mathbf{Q} = \mathbf{X}\mathbf{W}_Q, \ \mathbf{K} = \mathbf{X}\mathbf{W}_K, \ \mathbf{V} = \mathbf{X}\mathbf{W}_V \in \mathbb{R}^{t \times d_k}, \tag{1}$$

where $\mathbf{W}_Q, \mathbf{W}_K, \mathbf{W}_V \in \mathbb{R}^{d \times d_k}$ are parameterized weight matrices. The updated token embeddings are estimated by computing the self-attention matrix and multiplying it by the value matrix $\mathbf{V}$. The following equation represents the desired operation,

$$\mathrm{SA}(\mathbf{Q}, \mathbf{K}, \mathbf{V}) = \left( \frac{\mathbf{Q}.\mathbf{K}^\top}{\sqrt{d_k}} \right) \mathbf{V} \in \mathbb{R}^{t \times d_k}. \tag{2}$$

The self-attention, as mentioned earlier, is designed for a single head. For a multi-head self-attention module (MSA), we concatenate the individual heads. For $h$ heads, we have,

$$\mathrm{MSA}(\mathbf{Q}, \mathbf{K}, \mathbf{V}) = \left( \overset{h}{\underset{i=1}{\|}} \mathbf{H}_i \right) \mathbf{W}^\mathbf{O}, \tag{3}$$

where each head is estimated as $\mathbf{H_i} = \mathrm{SA}(\mathbf{Q}\mathbf{W_Q^i}, \mathbf{K}\mathbf{W_K^i}, \mathbf{V}\mathbf{W_V^i})$, and $\mathbf{W_Q^i}, \mathbf{W_K^i}, \mathbf{W_K^i}, \mathbf{W^O}$ are learnable projection matrices for $i^{th}$ head. Subsequently, the feature matrix is weighted by the attention matrix and fed into feedforward layers. The two linear transformations performed by the layers are presented as follows,

$$\mathrm{FFN}(\mathbf{X}) = \sigma(\mathbf{X}\mathbf{W_1} + \mathbf{b_1})\mathbf{W_2} + \mathbf{b_2}, \tag{4}$$

where $\sigma$ is an activation function (especially ReLU or GELU) and $\mathbf{W_1}, \mathbf{W_2}, \mathbf{b_1}, \mathbf{b_2}$ are learned weights and biases. Notably, we incorporate the positional information of input tokens by adding Positional encodings $\mathrm{PE}_i \in \mathrm{r}^d$ as follows,

$$z_i = \mathbf{X_i} + \mathbf{PE}_i. \tag{5}$$

The output of each layer is computed with residual connections and layer normalization, which is presented as follows,

$$\tilde{z}_i = \text{LayerNorm}(z_i + \text{MSA}(z_i, z_i, z_i))$$
$$\bar{z}_i = \text{LayerNorm}(\tilde{z}_i + \text{FFN}(\tilde{z}_i)). \tag{6}$$

The aforementioned Transformer architecture is delineated for a sequence of tokens. Precisely, for language tasks, the set of tokens $\mathbf{S}$ can be words or phrases. Similarly, for vision tasks, the $\mathbf{S}$ can be a sequence of image patches. For graph data, $\mathbf{S}$ can be a sequence of nodes, edges, or subgraphs.

### 2.2 Message Passing Graph Neural Network (MP-GNN)

The message passing framework (Gilmer et al., 2017) is the key component in a GNN layer. This comprises the following key functions.

- **Message** For every adjacent node pair $(i, j)$, the MSG function estimates messages between them as $m_{ij} = \text{MSG}(x_i, x_j)$

- **Aggregation** The messages are aggregated across the neighborhoods of the centering node. The following operation performs the neighborhood aggregation as $a_i = \text{AGG}(\{m_{ij}\}\forall j \in N(i))$.

- **Update** The aggregated messages are combined with the current feature of node $i$ to obtain the updated node feature as $x_i' = \text{UP}(x_i, a_i)$.

- **Readout** For the graph classification task, we require the representation of the entire graph. The embedding of the graph is computed by applying a readout function as $g_i = \text{READOUT}(X_i, A_i)$ for the $i^{th}$ graph.

### 2.3 Virtual Node (VN)

The concept of a Virtual Node was first introduced by (Pham et al., 2017). They proposed that the new node will be bidirectionally connected with every other node. The concept of VN catalyzes the research in the paradigm of graph neural networks (Battaglia et al., 2018; Gilmer et al., 2017). The addition of virtual nodes mitigates the Oversquashing phenomenon by enabling the shortest path distance to be either 1 or 2. The inclusion of VNs significantly improves the performance of graph classification tasks. (Hwang et al., 2022) offered the theoretical analysis on the effects of virtual nodes in the context of link prediction. Notably, (Southern et al., 2024) theoretically proved that effective resistance is reduced with the addition of virtual nodes in a graph. Recently, the analysis of virtual nodes has been connected with Graph Transformers (Cai et al., 2023; Rosenbluth et al., 2024).

## 3 Connection between `CLS` token and Virtual Node

In language and vision Transformers, a `CLS` token is prepended to the input sequence of tokens. Following the framework established by (Joshi, 2025), the self-attention mechanism operates on a complete graph constructed over all tokens. The introduction of the `CLS` token modifies the graph topology by adding a node with edges connecting to all other tokens in the sequence. Each node maintains a query vector $q_i$ and one key vector $k_i$. For each pair of tokens $(i, j)$, the attention score is computed by estimating the dot products $\langle q_i, k_j \rangle$ as depicted in the Figure 1. Token embeddings are subsequently updated through attention-weighted feature aggregation from neighboring tokens.

Notably, embedding of the `CLS` token encapsulates information aggregated from all input tokens, with its updated representation serving as input to the classification head. In graph neural networks, Virtual Nodes (VNs) exhibit analogous connectivity patterns, establishing connections to every node in the graph. Even for the bidirectional connections, the connectivity patterns remain unaffected. Similar to the `CLS` token, embeddings of VN are updated by aggregating features from all other nodes, though this aggregation may not necessarily employ attention weighting.

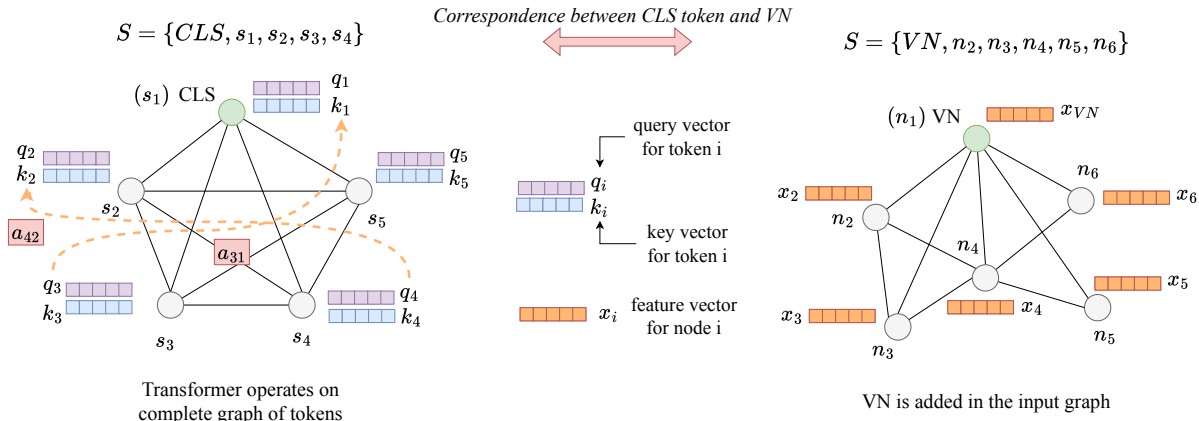

Figure 1: The structural similarity between `CLS` token and VN is presented. The number of nodes in the token graph and the input graph may not be equal. Still, both are connected to every other token or node in their respective graphs. `CLS` token aggregates the attention-weighted features from the remaining tokens. The attention weights are computed between the dot products between $q_i$ and $k_j$. On the contrary, the embedding of VN is updated by aggregating only the node features $x_i$.

The key insight is that both `CLS` tokens and VNs employ fundamentally similar feature update mechanisms. Both components maintain global connectivity within their respective graph structures and aggregate information from all other tokens or nodes. While the embeddings of `CLS` token are designed to generate representations for the classification of token sequences, VNs are employed to mitigate oversquashing in molecular graphs by reducing pairwise shortest path distances to at most 2 hops. Despite their distinct objectives, `CLS` tokens and VNs exhibit structural equivalence within their respective graphs. This observation suggests that embeddings of VNs may possess similar representational capacity for capturing global graph characteristics, warranting investigation into their effectiveness for graph classification tasks. This investigation will lead us to claim whether *"CLS token and VN are both sides of the same coin"*.

## 4 Experiments

### 4.1 Motivation

The primary function of the `CLS` token is to generate discriminative representations for classification tasks. In graph classification, graph-level representations are typically derived through global pooling operations, including Mean, Max, and Sum pooling. However, the potential of Virtual Node (VN) embeddings as graph representations has received limited attention. To address this gap, we propose utilizing the final VN embedding as a direct graph representation, which we term *VN-pooling*. We empirically evaluate this approach against established pooling techniques to assess its effectiveness in graph classification tasks.

### 4.2 Datasets

We performed experiments on six standard graph datasets from TUDatasets (Ivanov et al., 2019). The datasets are ENZYMES, MUTAG, PROTEINS, COLLAB, IMDB-BINARY, and REDDIT-BINARY. We also included two graph datasets from LRGB (Dwivedi et al., 2022), namely Peptides-func and Peptides-struct. These two datasets specifically face the problem of oversquashing, which requires long-range message passing.

### 4.3 Experimental Settings

We add exactly one Virtual Node to each graph for all eight datasets. The representation obtained from VN is used for predicting the graph label. GCN (Kipf & Welling, 2016) and GIN (Xu et al., 2018) are

Table 1: The performance of VN-pooling with three distinct initializations is compared against standard pooling methods across datasets. VN-pooling achieves competitive or superior performance relative to six conventional pooling techniques. Best test accuracies are highlighted in bold.

| GCN | | | | | | |
|---|---|---|---|---|---|---|
| Readout | ENZYMES | MUTAG | PROTEINS | COLLAB | IMDB-BINARY | REDDIT-BINARY |
| Mean | $25.00 \pm 8.35$ | $68.00 \pm 6.75$ | $65.71 \pm 5.43$ | $66.76 \pm 1.53$ | $64.50 \pm 4.30$ | $78.20 \pm 2.67$ |
| Sum | $16.83 \pm 3.37$ | $67.00 \pm 7.15$ | $70.80 \pm 4.49$ | $65.60 \pm 2.29$ | $51.30 \pm 4.42$ | $71.00 \pm 4.62$ |
| Max | $\mathbf{28.50 \pm 5.47}$ | $69.00 \pm 12.20$ | $70.09 \pm 4.28$ | $67.48 \pm 1.66$ | $62.60 \pm 2.88$ | $72.55 \pm 3.22$ |
| TopKPooling | $16.67 \pm 4.01$ | $53.00 \pm 10.33$ | $58.30 \pm 5.50$ | $52.56 \pm 1.58$ | $47.20 \pm 6.43$ | $59.85 \pm 5.21$ |
| SAGPooling | $20.00 \pm 5.33$ | $62.50 \pm 16.37$ | $55.54 \pm 6.23$ | $51.66 \pm 1.64$ | $50.00 \pm 4.97$ | $52.00 \pm 6.32$ |
| ASAPooling | $20.67 \pm 5.94$ | $66.00 \pm 6.58$ | $59.91 \pm 5.62$ | $58.80 \pm 2.19$ | $60.60 \pm 8.46$ | $64.50 \pm 2.77$ |
| VN-pooling-0 (Ours) | $25.33 \pm 5.82$ | $64.00 \pm 9.66$ | $\mathbf{71.88 \pm 2.57}$ | $67.94 \pm 2.25$ | $61.00 \pm 4.45$ | $78.35 \pm 1.40$ |
| VN-pooling-1 (Ours) | $26.83 \pm 5.58$ | $\mathbf{72.50 \pm 15.68}$ | $72.86 \pm 4.46$ | $66.78 \pm 1.99$ | $65.50 \pm 3.69$ | $\mathbf{78.80 \pm 2.69}$ |
| VN-pooling-$r$ (Ours) | $25.67 \pm 4.98$ | $70.00 \pm 11.30$ | $70.54 \pm 3.72$ | $\mathbf{68.54 \pm 1.45}$ | $\mathbf{66.50 \pm 3.69}$ | $77.50 \pm 4.61$ |
| GIN | | | | | | |
| Readout | ENZYMES | MUTAG | PROTEINS | COLLAB | IMDB-BINARY | REDDIT-BINARY |
| Mean | $\mathbf{27.17 \pm 5.67}$ | $\mathbf{68.00 \pm 7.89}$ | $62.77 \pm 3.23$ | $\mathbf{66.40 \pm 1.74}$ | $52.70 \pm 3.13$ | $53.50 \pm 4.91$ |
| Sum | $25.83 \pm 8.32$ | $66.50 \pm 8.18$ | $\mathbf{70.36 \pm 4.03}$ | $65.62 \pm 2.46$ | $47.20 \pm 4.13$ | $71.80 \pm 3.81$ |
| Max | $26.00 \pm 6.54$ | $67.00 \pm 6.75$ | $66.70 \pm 5.31$ | $60.16 \pm 2.66$ | $\mathbf{52.90 \pm 5.32}$ | $69.90 \pm 4.89$ |
| TopKPooling | $18.33 \pm 5.72$ | $62.00 \pm 10.06$ | $59.55 \pm 4.61$ | $56.86 \pm 2.64$ | $46.60 \pm 6.52$ | $53.80 \pm 6.11$ |
| SAGPooling | $15.67 \pm 5.34$ | $60.50 \pm 8.64$ | $57.23 \pm 4.20$ | $57.34 \pm 2.48$ | $49.50 \pm 6.64$ | $50.80 \pm 4.39$ |
| ASAPooling | $19.17 \pm 4.53$ | $64.50 \pm 11.17$ | $58.66 \pm 5.73$ | $55.00 \pm 2.30$ | $50.50 \pm 5.82$ | $51.65 \pm 6.01$ |
| VN-pooling-0 (Ours) | $14.50 \pm 5.39$ | $64.00 \pm 7.75$ | $57.59 \pm 5.43$ | $55.54 \pm 1.92$ | $49.60 \pm 4.43$ | $\mathbf{72.20 \pm 3.29}$ |
| VN-pooling-1 (Ours) | $16.17 \pm 2.73$ | $67.00 \pm 6.32$ | $56.96 \pm 7.50$ | $52.26 \pm 1.33$ | $48.50 \pm 3.75$ | $51.25 \pm 3.08$ |
| VN-pooling-$r$ (Ours) | $13.50 \pm 5.12$ | $57.00 \pm 8.23$ | $58.21 \pm 4.57$ | $52.32 \pm 1.46$ | $47.70 \pm 4.88$ | $48.35 \pm 3.67$ |

employed as backbone models. Following the standard protocol, we designed four convolutional layers and applied LayerNorm layers between the layers. The ReLU activation function is used, and the dropout rate is set to 0.50. The model parameters are optimized using the Adam optimizer, and the initial learning rate is set to 0.001. The weight decay is set to 1$e$-5 and the *ReduceLROnPlateau* scheduler is employed for faster convergence of training. For each dataset, we generate 10 random splits with 80%/**10**%/10% training, validation, and test graphs. The test accuracy with standard deviation is obtained by averaging over the splits for each dataset. For LRGB graphs, we considered standard data splits as mentioned in the (Dwivedi et al., 2022). The results are reported over 5 runs with different random seeds.

**Initializations of VN Embeddings.** We evaluate three distinct initialization strategies for VN embeddings: (1) zero initialization (VN-pooling-0), where all features are set to zero; (2) ones initialization (VN-pooling-1), where all features are initialized to one; and (3) random initialization (VN-pooling-$r$), where features are sampled from a standard normal distribution $\mathcal{N}(0, 1)$. For each dataset, we conduct separate experiments across all three initialization schemes to assess their impact on classification performance.

### 4.4 Baselines

We compare our method with commonly used pooling techniques like Mean, Max, and Sum. Additionally, we also considered three well-regarded advanced pooling techniques (1) TopKPooling (Gao & Ji, 2019), (2) SAGPooling (Lee et al., 2019), and (3) ASAPooling (Ranjan et al., 2020) to establish the efficacy of VN-based pooling operation.

### 4.5 Results & Observation

Tables 1 and 2 present the numerical results for graph classification across eight datasets using GCN and GIN as backbones. We compared our approach, VN-pooling, with six established pooling techniques for graph-level representation learning. The empirical results yield the following key findings:

1. The three schemes of VN-pooling demonstrate competitive or superior performance compared to conventional pooling methods across most experimental settings. Notably, both GCN and GIN

Table 2: Comparative performance of VN-pooling against standard pooling methods on Peptides datasets. Results are shown for both GCN and GIN architectures. For Peptides-func, higher values indicate better performance (AP metric), while for Peptides-struct, lower values indicate better performance (MAE metric). Best results are highlighted in bold.

| Readout/Backbone | PEPTIDES-FUNC | | PEPTIDES-STRUCT | |
|---|---|---|---|---|
| | GCN | GIN | GCN | GIN |
| Mean | $0.5961 \pm 0.0032$ | $0.6011 \pm 0.0028$ | $0.3544 \pm 0.0045$ | $0.3526 \pm 0.0038$ |
| Sum | $0.6138 \pm 0.0041$ | $0.6262 \pm 0.0037$ | $0.3013 \pm 0.0052$ | $0.2939 \pm 0.0048$ |
| Max | $0.4118 \pm 0.0056$ | $0.5123 \pm 0.0049$ | $0.3911 \pm 0.0061$ | $0.3740 \pm 0.0054$ |
| TopKPooling | $0.3092 \pm 0.0072$ | $0.2906 \pm 0.0068$ | $0.5634 \pm 0.0083$ | $0.5909 \pm 0.0091$ |
| SAGPooling | $0.3272 \pm 0.0064$ | $0.2419 \pm 0.0077$ | $0.5935 \pm 0.0088$ | $0.5362 \pm 0.0079$ |
| ASAPooling | $0.2881 \pm 0.0069$ | $0.2953 \pm 0.0071$ | $0.5706 \pm 0.0085$ | $0.5569 \pm 0.0082$ |
| VN-pooling-0 (Ours) | $\mathbf{0.6890 \pm 0.0018}$ | $0.6911 \pm 0.0021$ | $\mathbf{0.2496 \pm 0.0029}$ | $0.2610 \pm 0.0035$ |
| VN-pooling-1 (Ours) | $0.6731 \pm 0.0019$ | $\mathbf{0.6913 \pm 0.0017}$ | $0.2523 \pm 0.0031$ | $\mathbf{0.2543 \pm 0.0027}$ |
| VN-pooling-$r$ (Ours) | $0.6730 \pm 0.0022$ | $0.6836 \pm 0.0024$ | $0.2604 \pm 0.0033$ | $0.2573 \pm 0.0030$ |

with VN-pooling achieve the highest accuracy on REDDIT-BINARY, with VN-pooling outperforming standard techniques in 6 out of 12 experimental configurations in Table 1. For both Peptides datasets, VN-pooling consistently outperforms six contender pooling methods in every setting illustrated in Table 2. Furthermore, the three VN-pooling initialization techniques achieve optimal performance across multiple settings, highlighting the crucial role of initialization in determining the effectiveness of VN embeddings. These experimental results support our hypothesis that VNs exhibit structural similarities to `CLS` tokens in language and vision Transformers, serving as effective global representation aggregators.

2. While VN-pooling shows promise, it does not universally outperform traditional pooling methods across all datasets and architectures. We emphasize that our primary contribution lies in establishing the structural correspondence between the mechanisms of `CLS` token and VN, rather than claiming universal superiority of VN-pooling. The results demonstrate that VN embeddings can provide a viable alternative representation strategy, potentially benefiting practitioners depending on the specific downstream tasks and experimental configurations.

# 5 Conclusion

In this work, we establish a connection between `CLS` token and VN in the respective token graph and input graph. We observe that `CLS` token and VN possess structural similarity as both are connected to every other token or node in their respective graphs. We also pointed out that the embedding of `CLS` token is used to classify the input sequence for language and vision tasks. Conversely, VN is incorporated to counter the effect of oversquashing, but its embedding is not explored to solve the classification task. In experiments, we demonstrated the impact of VN-based pooling with three different initializations in graph classification tasks. Our results are compared to three standard and three advanced pooling techniques and VN-pooling achieved competitive performances.

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
