# OpenReview forum: "On Connection between $\texttt{CLS}$ Token and Virtual Node: Are they both sides of the same coin?"
_TMLR — Withdrawn by Authors_

### Review · Reviewer_nyi5 · 2025-11-09

**Summary Of Contributions:**

This paper draws a conceptual and empirical link between two mechanisms widely used in deep learning: the CLS token in Transformers and the Virtual Node (VN) in Graph Neural Networks. Both play a role in aggregating global information, but have traditionally been treated as separate design ideas.

The authors argue that these mechanisms are structurally equivalent, each introducing a globally connected node that collects information from all others, and that this equivalence has practical implications for how global representations are formed in graph models. To support this, they:

1. Describe the structural correspondence between CLS tokens and VNs when Transformers are viewed as operating on complete attention graphs.

2. Propose VN-pooling, which uses the learned VN embedding as a graph-level representation in the same way the CLS token is used in Transformers.

3. Evaluate VN-pooling on eight benchmark datasets (six from TUDatasets, two from LRGB) using GCN and GIN backbones, comparing it against both standard and hierarchical pooling methods.

4. Show that VN-pooling performs competitively or better on most datasets, especially those requiring long-range message passing.

Overall, the paper’s message is that CLS tokens and VNs can be viewed as two realizations of the same architectural idea: a single globally connected node acting as an information bottleneck and aggregation point.

**Audience:**

Yes

**Audience Explanation:**

The paper should appeal to readers working on graph representation learning, attention mechanisms, or model architecture analysis. The analogy between CLS tokens and VNs is clear and potentially useful, and the experimental findings could influence how future hybrid architectures are designed. Some members of TMLR’s audience would be interested, though the paper would be easier to follow if the background material were shortened and the theoretical motivation expanded.

**Broader Impact Concerns:**

This is a conceptual and empirical paper about architectural design and does not raise new societal or ethical concerns.

**Claims And Evidence:**

Yes

**Claims Explanation:**

Both of the paper’s main claims are supported with reasonable evidence. The analogy between CLS and VN is clearly presented in Figure 1 and explained in the text. Although the argument is mainly qualitative, it is consistent with existing work interpreting Transformers as message-passing networks.

The experimental evidence is solid. The setup is clear, covers a diverse set of datasets and models, and compares against a sensible set of baselines. The results are consistent with the claim that the VN embedding carries useful global information.

The main limitation is that the analysis is descriptive rather than formal. The equivalence is intuitive and visually clear, but not derived from shared update equations or representational properties.

**Requested Changes:**

Critical for acceptance:

1. Please clarify the theoretical framing. The current argument is mostly descriptive. A short formal derivation showing that self-attention and message passing with a VN share a common aggregation form would strengthen the main claim.

2. Add a small probing or interpretability analysis. For example, show how the VN embedding evolves across layers, or compare it directly with pooled representations. This would help demonstrate that the VN truly functions as a global aggregator.

Recommended but not critical:

3. Test VN-pooling on a Graph Transformer model to show that the analogy extends to architectures where both mechanisms already coexist.

4. Discuss initialization effects. The three VN initialization schemes behave differently; a short discussion or ablation would clarify why.

5. Streamline the preliminaries section and add brief intuitive summaries after formal definitions. Also, please use \citet for in-text citations (currently it seems \citep is used for all citations, i.e., all citations are currently parenthetical even when they should be in-text).

---

### Review · Reviewer_Zjvq · 2025-11-10

**Summary Of Contributions:**

This paper observes that the Transformer’s CLS token and a GNN’s Virtual Node (VN) play analogous roles: each is a single node connected to all other nodes (tokens) in the model’s underlying graph. Based on this insight, the authors propose to use the VN’s embedding as a graph-level representation for classification (in the same way that the CLS token’s embedding is used in Transformers).

 They implement this “VN-pooling” idea by adding one virtual node to each input graph, propagating features through standard GNN layers, and feeding the final VN embedding into a classifier. Experiments on eight benchmark graph datasets using two backbone GNNs (GCN and GIN) show that VN-pooling achieves accuracy that is often comparable to or better than standard global pooling methods (mean, sum, max pooling and three advanced pooling techniques).

Thus the paper’s main contributions as per the authors are:
(i) Identifying the formal structural correspondence between CLS tokens and virtual nodes.
(ii) Demonstrating empirically that leveraging the VN embedding as a global descriptor yields competitive graph classification performance.

**Additional Comments:**

None

**Audience:**

Yes

**Audience Explanation:**

The conceptual link between CLS token and VN in transformers and GNNs could interest researchers working on both attention-based models and graph representation learning. Using VN-Pooling as an alternate pooling strategy is also a simple yet nice idea. However, given the limited novelty, the absence of a deeper analysis, and the relatively narrow experimental validation, the paper’s appeal may be moderate rather than broad.

**Claims And Evidence:**

No

**Claims Explanation:**

While the paper offers an interesting conceptual analogy between the CLS token in Transformers and the Virtual Node (VN) in GNNs, the claims are not convincingly supported by empirical or theoretical evidence. The connection remains largely qualitative, with limited quantitative validation or formal analysis. Assertions such as the VN’s ability to mitigate oversquashing are not backed by targeted experiments, and the VN itself could suffer from oversquashing by design.

Furthermore, the experimental evaluation is narrow in scope,restricted to standard benchmarks without testing on more challenging datasets or including stronger baselines like Graph Transformers with CLS tokens. As a result, the evidence presented is insufficient to substantiate the paper’s broader claims or demonstrate clear, measurable advantages of the proposed approach.

**Requested Changes:**

(i) The current formulation of VNs assigns equal weight to all other nodes, which contrasts with the CLS token’s aggregation mechanism that relies on attention-based weighting. Using GAT as the backbone could better capture this analogy and provide empirical support for it.

(ii) To further strengthen the claims, the study should consider evaluating the approach on additional architectures or tasks—for example, incorporating a Graph Transformer baseline that includes its equivalent of a 'CLS' token or a model with a built-in global token. If such experiments are not feasible, a detailed discussion of these architectures as related work would be valuable.

(iii) Experiments using more experiments on more challenging ogbg datasets such as ogbg-ppa, ogbg-molpcba and ogbg-molhiv etc. would further showcase the strength of the pooling mechanism. Further, adding in some experiments on graph regression tasks as well could strengthen the experiments.

(iv) The authors repeatedly assert that the introduction of virtual nodes (VNs) helps mitigate oversquashing. However, by definition, oversquashing occurs when information from a large number of distant nodes is forced through a limited-dimensional message-passing channel, causing critical signals to be compressed or lost.

Given that a VN aggregates information from all nodes in the graph, one could argue that the VN’s embedding itself is highly susceptible to oversquashing, as it must encode the entire graph’s information into a single representation. The authors are encouraged to design quantitative experiments that explicitly demonstrate how the VN mechanism alleviates oversquashing, and to contrast this with potential oversquashing effects arising within the VN embedding itself.

(v) The results indicate that zero, one, and random initializations yield different accuracies. It would be informative to comment on this: for instance, why does initializing VN features to 1 sometimes perform better than 0? Discuss whether a trainable initial vector (like a learned parameter) might further improve performance.

(vi) To validate the claimed equivalence between the CLS token in Transformers and the Virtual Node (VN) in GNNs, the authors could include a text classification experiment where both mechanisms operate under comparable conditions.

For example, on the IMDB sentiment dataset, one could (a) fine-tune a Transformer using the [CLS] token for document-level prediction, and (b) build a token-level document graph (tokens as nodes, edges via co-occurrence or dependency links) and train an MPNN with a single VN used for classification.

Using shared pretrained token embeddings (e.g., from BERT) would enable a fair comparison and directly test whether the VN functions analogously to the CLS token in aggregating global context.

---

### Review · Reviewer_7fBv · 2025-11-14

**Summary Of Contributions:**

The paper draws a conceptual parallel between the CLS token used in Transformers and the Virtual Node (VN) in Graph Neural Networks (GNNs). The main claimed contributions are: (1) observing that both CLS tokens and VNs connect to all other nodes/tokens and aggregate features similarly; (2) proposing to use VN embeddings directly for graph classification instead of traditional pooling methods; (3) empirically evaluating VN-pooling with three initialization schemes on graph classification benchmarks.

Key strengths include clear empirical evaluation on 8 datasets showing competitive performance, reproducible setup, and a unifying perspective on two common techniques.

Key weaknesses:
- The core idea of using VN embeddings for graph classification was already published in 2017 (Pham et al.). Although they use the name "Column Networks", but it is essentially message passing GNNs published in parallel.
- The CLS-VN connection is obvious for those who work in in GNNs. No theoretical depth or novel architectural insights
- Mischaracterizes prior work by claiming VN embeddings were "not fully explored" when they were explicitly used for classification in 2017.

**Audience:**

Yes

**Audience Explanation:**

I put "Yes" assuming young students who have not studied the field deeply, but I doubt anyone serious enough would learn from the paper's claims. The paper's findings would have minimal impact on the field because: (1) the technical approach (using VN embeddings for classification) is not new; (2) no significant performance improvements are demonstrated; (3) the conceptual parallel lacks depth and actionable insights. The paper essentially confirms what was already known,  that is, VN embeddings can be used for graph classification.

**Broader Impact Concerns:**

No significant ethical concerns identified. This is research on graph neural network architectures without direct applications that raise ethical issues. The benchmarks used (molecular property prediction, graph classification) are standard academic datasets without sensitive information. No broader impact statement appears necessary for this work.

**Claims And Evidence:**

No

**Claims Explanation:**

The paper's central claim about bridging a "gap" in using VN embeddings for classification is factually incorrect: Pham et al. (2017) already demonstrated this exact approach. The experimental evidence shows only that VN-pooling is "competitive" with standard pooling methods, not superior, providing no compelling reason to adopt this approach. The paper's main claim of conceptual contribution, which draws the the CLS-VN parallel, is self-evident to the GNN community. There is no rigorous analysis or evidence of its utility beyond surface-level observation. The experimental validation is limited to showing comparable (not better) performance, failing to demonstrate any practical advantage of recognizing this connection.

**Requested Changes:**

- Address the novelty issue regarding Pham et al. (2017): The paper must acknowledge that using VN embeddings for graph classification was already demonstrated in the Virtual Column Network paper. Either clearly differentiate the contribution or pivot to a genuinely novel angle.
- Develop theoretical analysis of the CLS-VN connection and demonstrate how this insight can lead to improved architectures.
- Propose novel architectures leveraging the CLS-VN insight.
- Comprehensive experimental validation: Compare against Virtual Column Network (aka GNN with virtual nodes) (Pham et al. 2017) directly. Include more diverse benchmarks, especially those requiring long-range dependencies. Demonstrate clear advantages of the proposed approach

---

### Note · Authors · 2025-11-21

I have read and agree with the venue's withdrawal policy on behalf of myself and my co-authors.